# TOFU: Towards Obfuscated Federated Updates by Encoding Weight Updates into Gradients from Proxy Data

## Abstract

Advances in Federated Learning and an abundance of user data have enabled rich collaborative learning between multiple clients, without sharing user data. This is done via a central server that aggregates learning in the form of weight updates. However, this comes at the cost of repeated expensive communication between the clients and the server, and concerns about compromised user privacy. The inversion of gradients into the data that generated them is termed data leakage. Encryption techniques can be used to counter this leakage, but at added expense. To address these challenges of communication efficiency and privacy, we propose TOFU, a novel algorithm which generates proxy data that encodes the weight updates for each client in its gradients. Instead of weight updates, this proxy data is now shared. Since input data is far lower in dimensional complexity than weights, this encoding allows us to send much lesser data per communication round. Additionally, the proxy data resembles noise, and even perfect reconstruction from data leakage attacks would invert the decoded gradients into unrecognizable noise, enhancing privacy. We show that TOFU enables learning with less than $1\%$ and $7\%$ accuracy drops on MNIST and on CIFAR-10 datasets, respectively. This drop can be recovered via a few rounds of expensive encrypted gradient exchange. This enables us to learn to near-full accuracy in a federated setup, while being $4\times$ and $6.6\times$ more communication efficient than the standard Federated Averaging algorithm on MNIST and CIFAR-10, respectively.

## 1 Introduction

Federated learning is the regime in which many devices have access to localized data and communicate with each, other either directly or through a central node. The goal is to improve their learning abilities collaboratively, without sharing data. Here, we focus on the centralized setting, in which each device or 'client' learns on the data available to it and communicates the weight updates to a central node or 'server', which aggregates the updates it receives from all the clients. The server propagates the aggregated update back to each client, thus enabling collaborative learning from data available to all devices, without actually sharing the data. The abundance of user data has enabled rich complex learning. However, this comes at the cost of increased computational or communication costs between the clients and the server, and with increasing concerns about compromised user privacy. Privacy of user data is a growing concern, and standard federated averaging techniques have been shown to be vulnerable to data leakage by inverting gradients into the data that generated them (Zhu & Han, 2020; Geiping et al., 2020; Yin et al., 2021; Fowl et al., 2021; 2022). Gradients can be encrypted to preserve privacy, but incurs further communication overhead. (Bonawitz et al., 2017).

In this work, we focus on the communication between the clients and the server, a critical point for both communication and data leakage. Traditionally, in every communication round, each client shares its weight updates with the servers. To enable complex learning, the models are getting larger, growing to many millions of parameters (Simonyan & Zisserman, 2014; He et al., 2016). To put things in context, a VGG13 model has 9.4 million parameters, resulting in 36 MB of data being shared per communication round, per device. Since each device only has limited data, the number of rounds needed for the server to reach convergence are orders of magnitude more than those needed by the individual clients, further heightening the communication cost and opportunities for privacy leaks. This cost quickly grows prohibitive in resource constrained settings with limited bandwidth.

To address these concerns, we propose TOFU, a novel algorithm that works Towards Obfuscated Federated Updates, outlined pictorially in Figure 1. Here, each client generates synthetic proxy data whose combined gradient captures the weight update, and communicates this data instead of the weights. This mitigates two issues simultaneously. Data

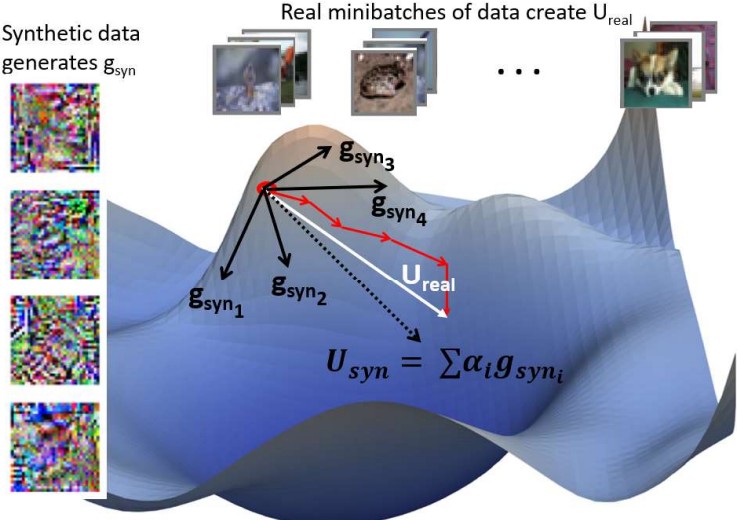

Figure 1: A pictorial representation of our encoding. The loss landscape shown in blue is taken from Li et al. (2017), with the starting point marked with a red circle. Each client learns on some minibatches of real data, shown on the top. The updates from these minibatches are marked with red arrows. The final weight update to be encoded and communicated, $U_{real}$, is shown in white. We construct a limited set of synthetic data that generates gradients $g_{syn}$ on the loss landscape, a weighted combination of which results in $U_{syn}$. The reconstruction algorithm optimizes these images and weights (denoted by $\alpha$) to maximize the cosine similarity between $U_{syn}$ and $U_{real}$. The synthetic images are visualized on the left and resemble noise, obfuscating user data.

is much lower dimensional than gradients. For context, CIFAR-10 images only have 3072 pixels, and we show that TOFU needs under 100 images to capture the weight updates well. Sending these images instead of the weight updates for VGG13 results in an order of magnitude reduction in communicated costs per round. Additionally, the synthetic data resembles noise, and existent techniques would invert the gradients to this noise rather than the true data, thus enhancing privacy. To further improve communication efficiency and encourage the synthetic images to differ from the true data distribution, we show that our method can approximate the gradient well even with images that are downsampled by $4\times$, or reduced to a single channel. The synthetic images are visualized in Figure 2.

Since our method approximates gradients to reduce communication costs and enhance privacy, it results in a slight accuracy drop. We exchange proxy data for most of the communication rounds, which are tolerant to noisy updates. Closer to convergence, updates are more precise and approximations are harmful. In these few communication rounds, we recover any accuracy drop by sharing the true full weight updates. In this phase, care needs to be taken to ensure privacy via expensive encryption techniques. Since this sensitive phase consists of far fewer communication rounds than the non-sensitive learning phase, the overheard resulting from this is countered by the communication efficiency achieved by sharing synthetic data instead of weight updates for most of the communication rounds. We show that we need only 3 and 15 full weight update rounds for MNIST and CIFAR-10, respectively, to recover any drop in accuracy.

This proposed hybrid approach provides both communication efficiency and privacy, without any loss in accuracy. We demonstrate TOFU on the CIFAR-10 dataset in single device setups and show that we can learn with $3\%$ accuracy drop while communicating $17\times$ lesser parameters on average. We extend this to a federated setup, with data distributed in an IID (Independent and Identically Distributed) fashion. We show that with a few additional rounds of full weight update, we can learn to accuracies comparable to FedAvg while achieving up to $4.6\times$ and $6.8\times$ better communication efficiency on MNIST and CIFAR-10, respectively. We emphasize that TOFU will result in increasing gains with increasing complexity of the models, since the size of weight updates will grow but the image size stays constant.

## 2 Background

(McMahan et al., 2017) pioneered the field of Federated Learning and proposed the first algorithm for distributed private training, called FedAvg, which serves as our baseline. Here, only weight updates are shared with the server, which aggregates updates from all clients and shares them back with each client. There are two research thrusts, that depend on whether the local data present at any client is distributed in an IID fashion or not. We focus on the IID

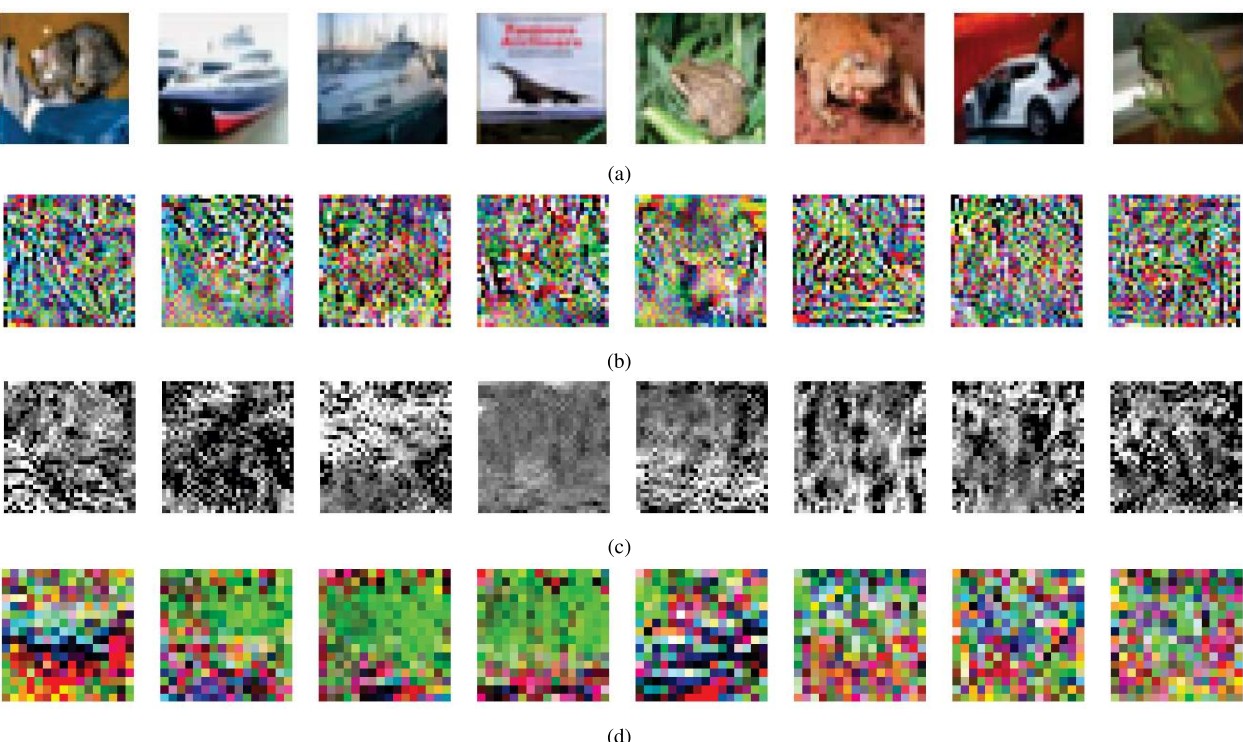

Figure 2: (a) True images sampled from the CIFAR-10 dataset. (b) Full-sized ($32 \times 32 \times 3$) synthetic images generated by TOFU. (c) Synthetic Images generated by downsampling the number of channels ($32 \times 32 \times 3$). (d) Synthetic Images generated by downsampling the height and width ($16 \times 16 \times 3$). The synthetic images encode the first round of weight exchange, corresponding to the learning from 200 minibatches.

setting in this work and direct readers to (Kairouz et al., 2019) for a better survey on non-IID methods. We focus on three key aspects in the IID setting: efficiency, privacy and accuracy, and discuss relevant works in each.

**Efficiency**  Federated learning has two key areas of inefficiency: communication cost, both from client to server (up-communication) and from server to client (down-communication), and computational cost. The most potential for impact comes with decreasing client to server communication (Kairouz et al., 2019). In our work, we target both up- and down-communication efficiency. Related works include quantization or sparsification of the weight updates (Konečný et al., 2016), (Horvath et al., 2019), (Basu et al., 2019), (Alistarh et al., 2016). While they significantly improve communication efficiency, there have been concerns raised (Kairouz et al., 2019) about their compatibility with secure aggregation (Bonawitz et al., 2017) and differential privacy techniques (Abadi et al., 2016). Our method can be thought of as an indirect compression, by encoding updates into proxy inputs. Our proxy images are amenable to encryption, and can potentially be further quantized, resulting in additional savings. In this work, we focus on showing that it is possible to encode gradients into synthetic fake-looking data and still enable learning. Other methods restrict the structure of updates, such as to a low rank or a sparse matrix (Konečný et al., 2016), or split the final network between the client and the server (He et al., 2020). We impose no constraints on learning, and focus on the standard case where each client has a synchronized model and equal accuracy on queries from any other client's dataset.

**Privacy**  Recent methods such as Inverting Gradients (IG) (Geiping et al., 2020) and Deep Leakage from Gradients (DLG) (Zhu & Han, 2020) have shown that gradients can be inverted into the training images that generated them, violating user data privacy. GradInversion (Yin et al., 2021) improved upon IG by introducing better fidelity metrics in the objective to recover images from federated scenarios with larger batch sizes, and more complex models and datasets such as ResNets and ImageNet. This is cause for concern, and we circumvent by showing that our proxy data looks like noise, and hence even perfect inversion by these techniques would only resemble noise. Fowl et al. (2021) show that altering the model architectures minimally allows the server to obtain user data without solving complex optimization problems. They also show that modifying only the larger linear layers can help recover user data. Methods to secure gradients from attack involve encryption and differential privacy techniques that add additional computational expense

(Bonawitz et al., 2017), (Abadi et al., 2016). These methods are compatible with our proxy data, should the need for extra encryption arise. Additionally, encrypting our proxy data will be less costly since standard encryption costs are proportional to the size of the vector being encrypted (Bonawitz et al., 2017).

**Accuracy** Efforts to increase accuracy often focus on variance reduced Stochastic Gradient Descent (SGD) (Karimireddy et al., 2019), (Yu et al., 2019) or on adaptive optimization and aggregation techniques (Reddi et al., 2020), (Karimireddy et al., 2020). Astraea (Duan et al., 2019) reschedules client participation based on the KL divergence of their data distribution in order to overcome data distribution imbalances to improve accuracy in federated settings. Our method is orthogonal and compatible with such techniques.

**Distillation** Recently, there has been interest in one-shot federated learning, wherein there is only one communication round. An approach that is similar to ours, called DOSFL (Zhou et al., 2020) focuses on this setting. It is based on the dataset distillation (Wang et al., 2018) technique, in which the entire dataset is distilled into synthetic data. DOSFL uses this to distill the local data of each client and share that for one-shot learning. There are a few key differences between our method for synthetic data generation and dataset distillation. We generate proxy data that aligns its gradients to a desired weight update, whereas dataset distillation optimizes data for accuracy after learning on it. Dataset Distillation shows very large drops in accuracy for CIFAR-10 dataset ($\sim$26%) versus our single device results (Section 4.1), which shows an average of 3% drop. DOSFL gets impressive results on MNIST, especially for a single round of communication but does not show results on larger datasets like CIFAR-10, presumably due to the significant drop in the baseline technique of dataset distillation. In parallel with the development of this work, (Cazenavette et al., 2022) came up with an extension of dataset distillation that precomputes training trajectories from an expert and saves them to guide the synthetic image creation. While the work overlaps with ours in concept, they focus on creating coresets to make training efficient, and we focus on improving communication efficiency in federated learning.

# 3 Methodology

This section outlines the desired properties of the synthetic dataset, the algorithm to create it, and the tradeoff between communication efficiency, privacy and accuracy.

## 3.1 Desiderata of the Synthetic Dataset

The generated synthetic dataset should have two properties - (a) it must be small in size in order to ensure communication efficiency and (b) it should not resemble the true data to ensure that data leakage attacks are unable to invert the proxy gradients into real data, thus ensuring privacy of the real data. We discuss these in more detail next.

**Communication Efficiency:** Input data is much lower in dimensional complexity than gradients (for instance, 3072 parameters per image in CIFAR-10 compared to 9.4 million parameters for sending VGG13 weight updates). This allows us to attain the first goal of efficient communication. We experimentally show that 64 images give us good results, which allows us to send $\sim$50$\times$ lesser data per communication round as compared to the weight updates of VGG13. For improving efficiency, we also experiment with images that have (a) the 3 channels reduced to a single channel, replicated across all 3 channels, and (b) the height and width reduced by 2$\times$, which is then upsampled with their nearest neighbor values as part of pre-processing. This adds a savings of 3$\times$ and 4$\times$, respectively.

**Enhanced Privacy from Data Leakage:** To attain the goal of privacy, we rely on the high dimensional, non-linear nature of neural networks to generate images that resemble noise to the human eye. We distill the weight update of a client after learning on many minibatches into a single minibatch of synthetic images. Combining weighted gradients is not the same as combining inputs, and we observe that condensing the learning from many images into a smaller set results in images that visually do not conform to the true data distribution. The resulting weighted gradient we generate is an approximation of the true weight update. This lossy compression buffers our gradient from data leakage as can be observed from the results of performing the Inverting Gradients (IG) attack, (Geiping et al., 2020)) on our images, shown in Figure 3. Even if IG attacks were to invert the images perfectly, the inverted images would still look like noise, circumventing data leakage. We employ some additional tricks to encourage obscurity of generated images. IG assumes availability of one-hot labels, or reconstructs one label per image. Instead, we use soft labels to further discourage reconstruction. Additionally, we weigh the gradients differently into the combined final gradient so that no true gradient is well represented. We distill the updates from a large number of minibatches (to the tune of

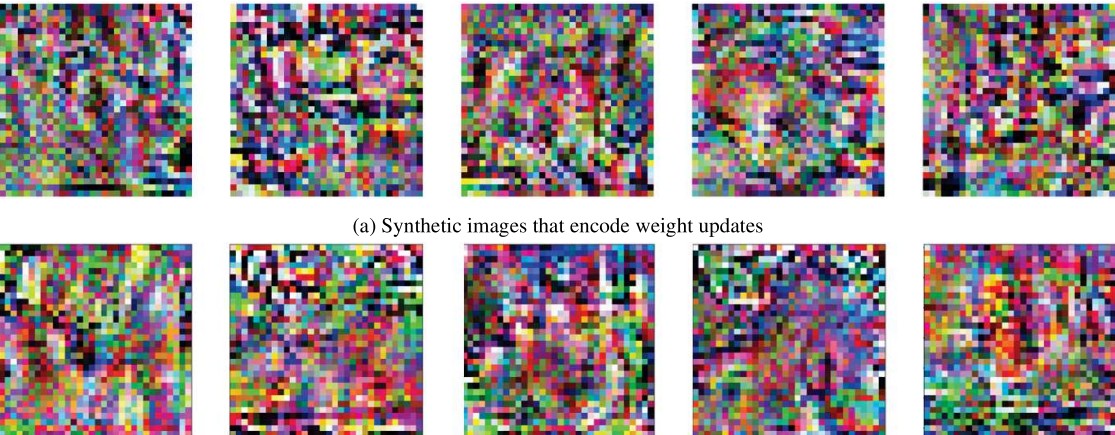

(a) Synthetic images that encode weight updates

(b) Images recovered by IG attack on synthetic images

Figure 3: The top row visualizes actual synthetic images ($x_{syn}$) generated by our algorithm. We show 5 randomly picked images from a set of 32 images encoding a weight update at the $200^{th}$ communication round of VGG13 on CIFAR-10, *Synfreq* = 1 epoch. The bottom row visualizes images recovered with the IG attack  Geiping et al. (2020) on the decoded weight update. Neither sets of images resemble CIFAR-10 images, visually obfuscating the user data and protecting it from leakage.

a whole epoch) down to just a handful of images. While this incurs an accuracy loss, it results in images that do not resemble the real data distribution at all. Additionally, forcing a downsampling prior on the images, in either channels or height and width, allows the image distribution to differ from the original one even more. These synthetic images are visualized in Figure 2 at the beginning of training. We also visualize the synthetic images generated by TOFU in the middle of the training process (epoch 200) and at maximum accuracy (epoch 400) in Supplementary section A.2.

**Tradeoffs:** The tradeoff cost associated is two-fold: a) the clients have added computational complexity to create the synthetic data and b) the communication rounds needed for convergence increase since we introduce some error in the weight updates. We emphasize that our method is better applied to use-cases where the clients have computational resources but are limited in communication bandwidth or cost. Furthermore, communication efficiency has been identified as the major efficiency bottleneck, with the potential for most impact (Kairouz et al., 2019). We account for the latter tradeoff of increased communication rounds when reporting our final efficiency ratios.

## 3.2 Creating the Synthetic Dataset

We now detail the algorithm that distills the change in model parameters into a synthetic dataset during training. Taking inspiration from the IG attack, we optimize synthetic data to align the resulting gradient direction with the true weight update. To formalize, let this true weight update attained after a client learns on its true data be referred to as $U_{real}$. We want to generate a synthetic dataset,

$$\mathcal{D}_{syn} = \{(x_{syn_i}, y_{syn_i}, \alpha_{syn_i}); \ i = 1...N\}$$

where $N$ is the number of images in the synthetic dataset. $x_{syn_i}$ and $y_{syn_i}$ refer to the $i^{th}$ image and soft label respectively. The goal of reconstruction is that the combined gradient obtained upon forward and back-propagating all $\{x_{syn_i}, y_{syn_i}\}$ is aligned to the true weight update, $U_{real}$. Each synthetic datapoint generates a single gradient direction, and with $N$ datapoints in our synthetic dataset, we generate $N$ different gradient directions. Traditionally, if we were to treat these $N$ images as a minibatch, we would average the $N$ gradients. However, we take a weighted average of the gradients, allowing us to span a larger space. We jointly optimize these weights, referred to as $\alpha_{syn_i}; \sum_i \alpha_{syn_i} = 1$, along with the images and soft label.

Next, we show that weighing the gradients of each image in the backward pass is the same as weighing the losses from each image on the forward pass, since derivative and summation can be interchanged. Formally, let $\theta$ be the weights of the model, and $\theta(x)$ the output of the model. Let the loss per synthetic datapoint, denoted by $L(\theta(x_{syn_i}), y_{syn_i})$, be weighted by the respective $\alpha_{syn_i}$ and summed into $L_{syn}$, the overall loss of the synthetic dataset. Let the resulting

gradient from backpropagating $L_{syn}$ be $U_{syn}$. Backpropagating $L_{syn}$ results in the desired weighted average of the individual gradients per datapoint, as shown:

$$L_{syn} = \sum_i \alpha_i L(\theta(x_{syn_i}), y_{syn_i}) \tag{1}$$

$$U_{syn} = \frac{\nabla L_{syn}}{\nabla \theta} = \sum_i \alpha_i \frac{\nabla L(\theta(x_{syn_i}), y_{syn_i})}{\nabla \theta} \tag{2}$$

Standard cross entropy loss is used to calculate gradients from both the true and the synthetic data. The synthetic data is optimized by minimizing the reconstruction loss, $R_{loss}$, which is the cosine similarity between the true update $U_{real}$, and the synthetic update $U_{syn}$.

$$R_{loss} = \left(1 - \frac{<U_{real} \cdot U_{syn}>}{\|U_{real}\|_2 \|U_{syn}\|_2}\right) \tag{3}$$

Since minimizing $R_{loss}$ only aligns the directions of gradients, we additionally send scaling values for each layer from $U_{real}$ to scale up $U_{syn}$. To avoid cluttering notation, we leave this out since it only adds extra parameters equal to the number of layers. In addition, we also use soft labels instead of the hard labels used in the real dataset. This provides more flexibility for the optimization algorithm to create a better alignment between synthetic and true updates, and discourages attacks like IG, which rely on one-hot labels. We use Adam (Kingma & Ba, 2015) to optimize the randomly initialized images to generate a gradient that aligns with the true weight updates. We use learning rates 0.1 for images, labels and $\alpha$s, for 1000 iterations, decayed by a factor of 0.1 at the $375^{th}$, $625^{th}$, and the $875^{th}$ iteration. For the downsampling experiments, since the network expects inputs of size $32 \times 32 \times 3$, we replicate the single channel along all dimension in the grayscale case, and perform nearest neighbor upsampling before feeding in the images for the case with reduced image size.

### 3.3 TOFU: The Federated Learning Algorithm

We now put all the parts together and describe how we utilize the synthesized dataset to enable communication efficient and private federated learning. All clients and the server have the same model initialization before the learning phase starts. Every client first trains on its private local data for a few minibatches and determines the true weight update, $U_{real}$, as the difference between the starting and ending weights. This true weight update is encoded using synthetic data ($\mathcal{D}_{syn}$) as described in Section 3.2. $\mathcal{D}_{syn}$ and then communicated in lieu of weight updates to the server. The server decodes the information by performing a single forward and backward pass to get the encoded weight update. The server repeats this for proxy data received from all clients and averages the decoded updates. To ensure efficiency during down-communication as well, the server encodes its own weight update due to aggregation into proxy data, and sends this back to all clients. The clients then update their local models by decoding this information. The process is repeated until convergence. This is summarized in Algorithm 3 in Supplementary section A.1.

### 3.4 Efficiency-Privacy-Accuracy Tradeoff

The weight update statistics change with accumulation over different number of batches and convergence progress. We now introduce the various hyperparameters that need to be tuned, and their effect on accuracy, privacy and efficiency.

**Number of Synthetic Datapoints (*Nimgs*)**: The size of the synthetic dataset transmitted per communication round has a direct impact on privacy, communication efficiency, and accuracy. While a larger synthetic dataset provides better accuracy as the encoding will be closer to the true weight update, the communication efficiency of the algorithm decreases since we have to communicate more data. We note that using 64-128 datapoints gives us the best empirical results. We see larger approximation errors with smaller number of datapoints, buffering us more against attacks.

**Synthesis Frequency (*Synfreq*)**: This denotes how many minibatches of weight updates should be accumulated by the client before communicating with the server. In FedAvg, this is usually one epoch. A very large *Synfreq* results in larger accuracy drops, since a large accumulated weight cannot be well represented by few synthetic images, but allows

for enhanced privacy due to larger approximations. However, it degrades efficiency since we have to communicate more often per epoch.

**Phases to Improve Accuracy** *(switch)* In the initial phase of learning, the gradient step is quite error tolerant since there is a strong direction of descent. For the **single device experiments**, after a few communication rounds for warm-up, we empirically see better results by scaling the learning rate by the reconstruction error. This is implemented by scaling $U_{syn}$ by $(1\text{-}R_{loss})$, capturing the cosine similarity between the true and the synthetic update. This enables small steps to be taken if the synthetic data could not approximate the true update well. For single-device experiments, we switch from warm-up to this scaled learning rate phase at 200 communication rounds, and referred to it as $switch_1$. In the **Federated setting**, we did not notice much improvement from this switch empirically, and thus leave it out for simplicity. However, we note that two sets of encoding are required now, for both up-communication and down-communication, and hence, we see more accuracy drop than the single device case. To counter this, we end with a few communication rounds of full weight update exchange to regain any accuracy loss. To ensure privacy, we recommend expensive encryption of the weight updates here. Since it consists of very few rounds (under 15), we do not sacrifice efficiency. We call the communication round where we switch to this final phase as $switch_2$ and mark it by a star in the learning curves shown in Supplementary section A.3, and mention them in the corresponding hyperparameter sections. The efficiency savings we report take into account these rounds of expensive full gradient exchange as well. To summarize, for single device experiments, we have a brief warm-up phase of a few hundred communication rounds, and then switch to scaling learning rate by reconstruction error. For the federated setup, we do not scale the learning rate by communication round, and instead have a brief full weight update exchange phase of up to 15 communication rounds at the end of training.

**Experimenting with Image sizes**: We want to encourage synthetic data to resemble the true data distribution even lesser, while relying on our optimization algorithm to tune them to get a good match between the target gradient and the synthetic gradient. To do this, and to further enhance efficiency, we experiment with enforcing two downsampling priors on the synthetic images. The CIFAR-10 images are of size $32 \times 32 \times 3$, and this is also the size of our synthetic data. In one experiment, we downsample the number of channels in synthetic images from 3 to 1. The single channel is replicated across the 3 dimensions before being fed into the network for synthetic gradient calculation. In the other experiment, the synthetic images are sized $16 \times 16 \times 3$ with the height and width upsampled to the correct size by nearest neighbor upsampling before being fed into the network. We show a comparison of what the true images, the synthetic full size images, the grayscale images, and the downsampled images look like in Figure 2. In Section 4.1, we show that we can successfully learn with all three kinds of synthetic images with an average accuracy drop of 3% for full sized synthetic images, 5% for single-channeled images and 3.5% for images with width and length halved.

# 4 Experimental Results and Discussion

In this section, we first demonstrate TOFU on a single device setup to show that privacy preserved learning with only synthetic data is possible. This setup can be thought of as a federated setup with only 1 client and no down-communication. We initialize two copies of the same network with the same weights. Network 1 represents the client, and learns on real data for *Synfreq* number of batches, generating *Nimgs* number of synthetic datapoints to send to Network 2, which emulates the server. Network 2 only learns on the synthetic data. Post communication, both the networks have the same weights since Network 1 knows how the synthetic data is going to update Network 2 and resets its own weights accordingly. For the single device experiments, we focus on getting the maximum accuracy from purely synthetic data, and hence we do not employ the final phase of full weight update exchanges. We first show the results of learning with similarly sized images as the real data, and then introduce priors that recreate the gradient via single channel images and images with width and height downsampled by 2.

We then extend it to multiple clients in a federated setup. This has two encoding phases, the first carried out by each client to transmit their updates to the server (up-communication), and the second carried out by the server after aggregation from all clients (down-communication). Down-communication ensures that the weights of all clients and the server remain in sync after the end of each communication round. The experiments for Federated setup include a few rounds of full gradient exchange at the end in order to circumvent any accuracy drop, and the emphasis in these experiments is on efficiency. All learning curves are shown in Supplementary section A.3.

| Synthetic Image Size | Original 32×32×3 | | Grayscale 32×32×1 | | Downsampled 16×16×3 | |
|---|---|---|---|---|---|---|
| | Max Acc (%) | Comm Eff | Max Acc (%) | Comm Eff | Max Acc (%) | Comm Eff |
| Baseline Accuracy without using synthetic data: 88.6% | | | | | | |
| *Nimgs* | Varying Nimgs @Synfreq = 200 | | | | | |
| 32 | 84.05 | 32× | 81.24 | 97× | 82.61 | 121× |
| 64 | 85.78 | 18× | 83.39 | 45× | 84.14 | 60× |
| 96 | 86.05 | 10× | 83.63 | 31× | 85.29 | 46× |
| 128 | **86.81** | **8×** | **84.09** | **23×** | **85.74** | **30×** |
| *Synfreq* | Varying Synfreq @Nimgs = 64 | | | | | |
| 50 | 85.22 | 16× | 82 | 55× | 83.57 | 75× |
| 100 | 84.73 | 16× | 82.73 | 45× | 84.18 | 64× |
| 200 | **85.78** | **18×** | **83.39** | **45×** | 84.14 | 60× |
| 400 | 85.76 | 15× | 83.19 | 47× | 84.61 | 64× |
| 1 epoch | 84.79 | 19× | 81.3 | 51× | **84.62** | **64×** |

Table 1: Single Device accuracies and efficiency ratios for a VGG13 model, CIFAR-10 dataset on synthetic data. Baseline accuracy for learning on real data with the same hyperparameters is 88.6%, as shown in grey. The best accuracy setting for each set of experiments for using only synthetic data is highlighted in bold. Communication rounds required to reach maximum accuracy are shown in Supplementary section A.3.2. The network is trained with a batch-size of 64, with 782 minibatches making up an epoch.

## 4.1 Single Device Experiments

**Setup** We demonstrate our results on the CIFAR-10 dataset. It comprises of 50,000 training samples and 10,000 validation samples of 10 classes each. For all experiments, we use a VGG-13 (Simonyan & Zisserman, 2014) network. We use an SGD optimizer with learning rates 0.02, decayed by 0.2 at the $250^{th}$ and $400^{th}$ epochs, with a mini-batch size of 64 for a total of 500 epochs. The maximum baseline accuracy we achieve by training on real data is 88.6%. In this section all non-baseline results are shown for learning on purely synthetic data, with varying *Synfreq*, *Nimgs* and synthetic image sizes. More details are shown in Supplementary section A.3. The results are tabulated in Table 1. The efficiency shown is calculated as the ratio of parameters sent with synthetic data versus sending full gradients for the corresponding $Synfreq$ and $Nimgs$ setting. We perform the warm up phase for all experiments for 200 communication rounds and then start to scale the learning rate by the reconstruction error. The hyperparameters are tuned for the case of *Synfreq*= 200 and *Nimgs*= 64 and are held constant for all the other single device experiments. We show additional results for MNIST dataset in Supplementary section A.3.2.

**Efficiency Calculation** The number of parameters of each synthetic image is 3072 ($32 \times 32 \times 3$) if no prior is enforced on the image, 1024 ($32 \times 32 \times 1$) if single-channeled images are synthesized, and 768 ($16 \times 16 \times 3$) if downsampled images are synthesized. The size of each datapoint (image, label, $\alpha$ trio) in the synthetic dataset $\mathcal{D}_{syn}$ is the sum of the size of the synthetic image + 10 (number of soft labels) + 1 ($\alpha$ per image). Hence the total size of the synthetic dataset ($D_{size}$) is the size of each data point multiplied with *Nimgs*. The communication efficiency ($\eta$) is then calculated as:

$$\eta = \frac{\text{Number of Model Parameters} * \text{Number of Baseline Communication Rounds}}{D_{size} * \text{Number of Synthetic Communication Rounds}}$$

**Varying *Nimgs*** Table 1 shows that increasing the synthetic dataset size improves accuracy, but reduces communication efficiency as we need to send more parameters per communication round. We achieve good accuracies by learning on only synthetic data, with an average accuracy drop of $3\%, 5.5\%$ and $4\%$ for synthetic images of original size, grayscale images, and downscaled respectively across all considered *Nimgs*. For further experiments, we fix *Nimgs* = 64 as a good trade-off point. The exchange frequency for both the baseline and the synthetic case for all *Nimgs* is set as 200. Baseline training reaches full accuracy sooner than learning with synthetic data as expected, but even after accounting for that, we are able to achieve upto 121× more communication efficiency. The communication

rounds vary between experiments and are shown in Appendix section A.3.2. We also show the corresponding results for MNIST in Appendix section A.3.2.

**Varying *Synfreq***   Here we show that whether we generate synthetic images to match a gradient as often as 50 minibatches or as late as once an epoch (updates from 782 minibatches), we can converge to a reasonable accuracy. All simulations take similar number of epochs to converge, and are able to converge to very similar accuracies. In Table 1, we compare efficiency when the synthetic images are being communicated instead of the gradient, and we assume that both of these are communicated as often as the mentioned *Synfreq*. However, as we see in the federated setup in the next section, the frequency of communication can vary between the synthetic data and the real data. In those cases, a lower *Synfreq* will result in a requirement for more communication rounds and hence achieve lesser communication savings. The corresponding rounds for convergence for all experiments are mentioned in Supplementary section A.3.2.

**Forcing a Prior on Synthetic Image Sizes**   Here, we experiment with enforcing images to not follow the same distribution as the real dataset by constraining their size. In the first experiment, the synthetic images are constrained to have a single channel (results in column 2 of Table 1) . In the second experiment, we constrain images to be half the width and height (results in column 3 of Table 1). To be compatible with expected image size before being fed into the network for gradient calculation, the grayscale images are duplicated across the 3 channels and the smaller images are upsampled to the correct size by copying the nearest neighbors pixel value. The results show that we get very low drop in accuracies from full sized images. The average accuracy drop from full sized synthetic images to images with height and width halved is only $0.5\%$ acrosss all experiments , and $2\%$ to grayscale images. However we can see that we get approximately $4\times$ and $3\times$ improvement in communication efficiency as a result of reducing the number of parameters to be communicated or learned per image.

**Discussion**   We successfully show that synthetic data can be used to learn, with small accuracy drops for CIFAR-10, using only synthetic data. This drop is later recovered in the federated setup. For full size images, the average accuracy drop from baseline is $3\%$ at $17\times$ communication efficiency. For grayscale images, we get an average accuracy drop of $5\%$ at $49\times$ communication efficiency and $3.5\%$ drop at $65\times$ savings for images with width and height halved. We also wish to mention reiterate that the larger the networks, the more the savings that can be achieved with our method, since the size of updates will increase but the size of images remains constant.

## 4.2   Federated Learning Experiments

**Setup**   We now discuss the federated experiments conducted on 5 and 10 clients for CIFAR-10 and MNIST, respectively, with an IID distribution of data. This results in each client having 157 minibatches of size 64 image-label pairs for CIFAR-10 and 97 minibatches of size 64 for MNIST. We compare with FedAvg (McMahan et al., 2017) as our baseline, with exchanges happening once per epoch. We assume a participation rate of 1. The results are shown in Table 2 for MNIST and Table 3 for CIFAR-10.

**Efficiency Calculation**   In the case where full gradients are exchanged at the end in order to recover the accuracy, the communication efficiency ($\eta$) is calculated by including the parameters used for exchanging the full gradients ($P_{full}$):

$$P_{full} = (\text{Number of Model Parameters} * \text{Number of Full Gradient Exchanges})$$

$$\eta = \frac{\text{Number of Model Parameters} * \text{Number of Baseline Communication Rounds}}{(D_{size} * \text{Number of Synthetic Communication Rounds}) + P_{full}}$$

**Varying *Synfreq* and *Nimgs***   We get the best results at a *Synfreq* equal to one local epoch for both datasets, similar to FedAvg. We note that there is more variation in efficiency with varying *Synfreq* settings than single device experiments. That is because in this case, our Federated baseline is fixed at a *Synfreq* of 1 epoch, irrespective of what the *Synfreq* of synthetic images is. Increasing the synthetic dataset size (*Nimgs*) results in better or comparable accuracy but sends more parameters per communication round. We get best accuracy for *Nimgs* = 96 for both datasets, seeing a drop of $2\%$ and $12\%$ for MNIST and CIFAR-10, with $3.6\times$ and $14.7\times$ communication efficiency, respectively. We note that the drop is more than what we see in single device, since each round incurs approximations from both up and down-communications. We show that we can recover this drop almost completely in just 3 communication rounds for

| MNIST, LeNet5, 10 Clients with IID distribution | | | | |
|---|---|---|---|---|
| | Using Only Synthetic Data | | + 3 Additional Rounds of FedAvg | |
| | Max. Acc. (%) | Comm. Eff. | Max Acc. (%) | Comm. Eff. |
| FEDAVG | 98.91 | 1.0× | - | - |
| *Nimgs* | *Synfreq* = 1 local epoch | | | |
| 32 | 95.39 | 6.9× | 98.06 | 6.6× |
| 64 | 96.06 | 4.2× | 98.14 | 4.1× |
| **96** | **96.77** | **3.6×** | **98.01** | **3.5×** |
| 128 | 95.23 | 4.2× | 98.07 | 4.1× |
| *Synfreq* | *Nimgs* = 64 | | | |
| 25 | 92.00 | 3.4× | 98.29 | 3.3× |
| 50 | 91.71 | 7.9× | 98.19 | 7.5× |
| **1 epoch** | **96.06** | **4.2×** | **98.14** | **4.1×** |
| 2 epochs | 95.52 | 3.1× | 97.91 | 3.1× |

Table 2: Accuracy and efficiency of the federated platform on MNIST. The baseline of FedAvg is shown in grey. The best accuracy setting for each set of experiments for using only synthetic data is highlighted in bold.

MNIST and 15 rounds in CIFAR-10, still allowing for $3 - 6.6\times$ communication efficiency for MNIST and $5.4 - 8.8\times$ for CIFAR-10, as seen in the last columns of Tables 2 and 3. Additionally, we realize that showing maximum accuracy might skew communication efficiency in our benefit, since FedAvg reaches higher accuracy, which will naturally take more communication rounds. To account for this, we also report communication efficiency at iso-accuracy in Tables 7 and 8 for MNIST and CIFAR-10, respectively in Supplementary section A.3. When we consider 72% as the baseline accuracy for CIFAR-10, our efficiency of $5.4 - 8.8\times$ reduces to $2.4 - 4.7\times$. This shows that our method is still more efficient under stricter evaluation conditions. We found that the combination of *Synfreq*=1 epoch and *Nimgs*=64 provides a good trade-off point between communication efficiency and accuracy drop.

**Enforcing a Prior on Synthetic Image Sizes**   We also experimented with restricting the optimization to generate single-channeled images in the federated setup. Table 3 shows that generating 1-channeled images instead of 3-channeled images provided communication efficiency at the cost of accuracy. Using only synthetic images provided a communication efficiency of $46 - 117\times$, however, with an accuracy drop of $17 - 25\%$ from the FedAvg baseline. However, this drop can be reduced to lesser than 1% with 15 additional rounds of full weight update exchange, while still achieving communication efficiency of $8.9 - 10.1\times$.

**Cost of Full Weight Update Exchange**   TOFU shares full weight updates for the last few epochs to regain full accuracy, which need to be encrypted to ensure privacy. For a conservative estimate while ensuring privacy, we assume that we need to encrypt all parameters sent during all communication rounds for both methods, including synthetic data and full weight updates. Secure aggregation (Bonawitz et al., 2017), a commonly used protocol, shows that the communication cost is $\mathcal{O}(n + k)$ for the client and $\mathcal{O}(nk + n^2)$ for the server, where $k$ is the dimension of the vector being encrypted and $n$ is the number of clients. Comparing the encryption cost between FedAvg and TOFU for the same number of clients reduces to a ratio of the total parameters sent. This means that encryption retains the efficiency benefits of our method. The results show that TOFU can learn both MNIST and CIFAR-10, distributed in an IID setup, with an average of $\sim 4.6\times$ and $\sim 6.8\times$ communication efficiency and less than an 1% average accuracy drop.

## 5   Conclusion

In the standard federated learning algorithm, clients carry out local learning on their private datasets for some mini-batches, and communicate their weight updates to a central server. The central server aggregates the weight updates

| CIFAR-10, VGG13, 5 Clients with IID distribution | | | | |
|---|---|---|---|---|
| | | Using Only Synthetic Data | | + 15 Additional Rounds of FedAvg | |
| | | Max. Acc. (%) | Comm. Eff. | Max Acc. (%) | Comm. Eff. |
| **FEDAVG** | | **88.73** | **1.0×** | **-** | **-** |
| 3 Channel Synthetic Images | *Nimgs* | *Synfreq* = 1 local epoch | | | |
| | 32 | 67.12 | 44.9× | 87.29 | 8.8× |
| | 64 | 75.00 | 19.8× | 88.30 | 7.1× |
| | **96** | **76.02** | **14.7×** | **88.39** | **6.3×** |
| | 128 | 76.00 | 10.6× | 87.86 | 5.4× |
| | *Synfreq* | *Nimgs* = 64 | | | |
| | 50 | 63.03 | 18.0× | 87.69 | 6.9× |
| | 100 | 70.07 | 14.5× | 87.26 | 6.3× |
| | **1 epoch** | **75.00** | **19.8×** | **88.30** | **7.1×** |
| 1 Channel Synthetic Images | *Nimgs* | *Synfreq* = 1 local epoch | | | |
| | 32 | 63.26 | 117.0× | 87.17 | 10.1× |
| | 64 | 69.04 | 58.9× | 87.34 | 9.3× |
| | **96** | **71.15** | **39.3×** | **87.60** | **8.6×** |
| | 128 | 71.04 | 46.1× | 86.88 | 8.9× |
| | *Synfreq* | *Nimgs* = 64 | | | |
| | 50 | 54.23 | 47.2× | 87.57 | 8.9× |
| | 100 | 65.08 | 47.2× | 87.38 | 8.9× |
| | **1 epoch** | **69.04** | **58.9×** | **87.34** | **9.3×** |

Table 3: Accuracy and efficiency on the federated platform on CIFAR-10. The baseline of FedAvg is shown in grey. Results for additional 5 & 10 rounds are in Table 9, Supplementary section A.3.4. The best accuracy setting for each set of experiments for using only synthetic data is highlighted in bold.

received from all clients, and communicates this update back to all clients. There are two major bottlenecks in this procedure; it is communication inefficient and it is shown that gradient and weight updates can be inverted into the data that generated them, violating user privacy. In this work, we introduce TOFU, a federated learning algorithm for communication efficiency and to enhance protection against data leakage via gradients. We encode the weight updates to be communicated into a weighted summation of the gradients of a much smaller set of proxy data. The proxy data resembles noise and thus even perfect inversion from data leakage attacks will result in revealing this noise rather than user data. Additionally, data is far lower in dimensional complexity than gradients, improving communication efficiency. We also show that this proxy data can be downsampled in size from the original data that generated the target gradients without much drop in accuracy, thus being even more efficient. Since proxy data only approximates gradients, we observe a small drop in accuracy when learning only from this synthetic data. We show that the accuracy can be recovered by a very few communication rounds of full weight updates. To ensure privacy in this phase, we recommend encrypting the updates. Since these rounds are very few in comparison to the number of rounds where we exchange synthetic data, we are still able to maintain communication efficiency. We show that we can learn the MNIST dataset, distributed between 10 clients and the CIFAR-10 dataset, distributed between 5 clients to accuracies comparable to FedAvg, with an average of $\sim$4.6× and $\sim$6.8× communication efficiency and less than an average 1% accuracy drop, respectively. Availability of more data and compute capabilities has encouraged network sizes to grow. Since input data usually is of fixed dimensions, the communication efficiency advantages of TOFU are expected to grow with network size.

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
