# OpenReview forum: "TOFU: Towards Obfuscated Federated Updates by Encoding Weight Updates into Gradients from Proxy Data"
_TMLR — Rejected by TMLR_

### Review · Reviewer_7ihT · 2023-01-13

**Summary Of Contributions:**

The papers proposes a new federated learning scheme to reduce the communication cost by generating synthetic data. Specifically, to generate such data, it uses a loss function to let the generated synthetic data's gradient loss matches the model updates. To get a better matching performance, it further utilizes the soft label in stead of hard label and learn a parameter alpha to determine the importance of each samples. The experimental results show, for one client, the proposed method could still maintain similar performance (around 2-3% drop) with over 5X speed up in terms of number of communication cost. For multiple clients in the IID data distribution, it still suffer from around 10% performance loss when only using synthetic data.

**Audience:**

Yes

**Broader Impact Concerns:**

I don't see any ethical concerns in this paper.

**Claims And Evidence:**

No

**Requested Changes:**

Please refer to weaknesses.

**Strengths And Weaknesses:**

Strengths:
1. The proposed new scheme is interesting and novel. It is also interesting to apply data condensation into federated learning framework for a better communication efficiency.
2. The experimental results on one client looks promising. However, I would like to see more experimental results on more practical settings such as non-IID data.


Weaknesses:
1. The proposed method should include some formal proofs to show it still keeps the clients data's confidentiality. By directly sharing the data with the sever, it is necessary to show the synthetic data won't include data information. Although the generated synthetic data seems that it doesn't include the personal data, it is not enough to justify the confidentiality. In short, there is no enough evidence to show the privacy requirement in FL.
2. The experiments are conducted all on the IID datasets. It is necessary to show the proposed method's performance on non-IID distribution as well.

---

> ### Author Response · Authors · 2023-02-10
> **Response to Reviewer 7ihT**
>
>
> We would like to thank the reviewer for the encouragement and the feedback provided. Although we do not show any formal proofs of privacy, we perform a lot of additional experiments to strengthen the confidentiality claim. These have been added to the supplementary material in section A.4. We have also included other attacks that are able to decipher dataset information from the gradients when TOFU is not used, but  fail when TOFU is used. Here, we provide a summary of the new experiments to empirically reinforce the resulting data obfuscation from our method.
>
> &nbsp;
>
> 1.  In Figure 10, we show that for the same settings, Inverting Gradients (IG)  is able to recover user data when true gradients are shown, and fails to recover it when TOFU is used.
>
> &nbsp;
>
> 2.  In Figure 11, we show traditional IG attack and then modify it to work with soft labels. We show that both settings fail to reveal data.
>
> &nbsp;
>
> 3.  In Fig12, we start with the very basic case of a single true image’s gradient encoded into a single fake image. In this case, IG is able to recover the true gradient as expected, since there is no condensation happening here, and no role for spanning ratios and soft labels. But as we increase batchsize in Figure 12, Nimgs in Figure 13, and Synfreq in Fig 14, TOFU’s privacy claim holds and IG fails to recover data.
>
> &nbsp;
>
> 4.  In Figure 15, we show 4 attacks in total: IG, DLG, GradInversion and GIAS. We show that these attacks recover user data on true gradients, and fail to recover data with synthetic gradients produced by TOFU. We also show that these attacks converge to the same noisy looking data, to show that it is indeed our method that induces data obfuscation rather than a failing of these attacks.
>
> &nbsp;
>
> We hope that these additional results count towards strengthening our claim on privacy.

---

### Review · Reviewer_v217 · 2023-01-13

**Summary Of Contributions:**

The authors propose a novel method, TOFU, to address two common challenges in Federated Learning (FL) pipelines: (1) expensive client to server communication of model updates and (2) privacy concerns with gradient inversion attacks allowing to recover (a part of the) local training samples.

The main idea is to encode the model update (delta between the server model and the model after training for say one epoch locally) as a weighted sum of gradients of mock images that resemble noise. Once a client has designed the mock samples to encode their model update, they send the mock samples to the server, and the server is able to reconstruct (approximately) the model update by computing the weighted sum of the gradients of the mock samples with respect to the previous central model.

The authors claim two main benefits of TOFU
- Reduced communication cost (4 to 6x on FL LEAF benchmarks)
- Enhanced privacy (Figure 3 in particular shows that the IG attack (gradient inversion) fails to recover the original images.


**Audience:**

Yes

**Broader Impact Concerns:**

None.

**Claims And Evidence:**

No

**Requested Changes:**

The main requested changes are highlighted in the "Strengths and Weaknesses" section, namely:
- [high] Privacy claims: sanity check, push attack setup and redesign attack.
- [medium] Communication reduction claim: elaborate on compute and full model updates.
- [small] Typo (nit) "communicate with each, other" first sentence of introduction.


**Strengths And Weaknesses:**

Let us address the two main challenges raised by the authors in FL:

**Communication**

Here, the claim is matched with empirical evidence mentioned in the abstract and in Tables 1, 2 and 3. However, we would like to point important points that would deserve clarification:
- Compute: with TOFU, the authors implicitly trade off less communication for more compute: (1) on the client side, since each client has to perform approximately a thousand steps of gradient descent to design the mock samples (end of Section 3.2) after performing the local standard training and (2) on the server side, since the server has to recompute the gradients of all mock samples (roughly 64) coming from all the clients participating in the round (realistically, a few hundred rather than the proposed 5 or 10 clients of Section 4.2). It would strengthen the impact of the paper if the authors could clearly present and benchmark the additional compute needed, in particular on the client side. Indeed, the ultimate metric in FL is the wall time needed to reach a certain model utility or accuracy. If client compute time doubles, then wall time to accuracy would essentially double.
- Full model updates: the authors mention the necessity to still send over full model updates without using TOFU (a few of them over the course of the training), especially as model gets closer to convergence. Could the authors please elaborate on this point by providing loss curves (flagging when full model updates were performed) to help the reader quantify how much we need to recover when using only TOFU? It might be indeed that most of the loss incurred by TOFU is recovered with the full model updates, hence it would be important to substantiate this claim more.

**Privacy**

This experiments would deserve more clarification and investigation from the authors to claim privacy protection. The high-level concern is the following. The server is able to recover (a good approximate of) the individual model update of each client (as demonstrated by the decent accuracies reached after training, assuming non-TOFU full model updates are not a concern as highlighted above). One might think that there is therefore enough information in the individual model updates to recover the *original*, non-mock samples (or at least some of them), otherwise it would be impossible to learn a model performing well on original images. In other words, it is not because a single generic gradient inversion attack fails (as demonstrated in Figure 3) that the same attack with more aggressive setups or another, slightly modified gradient inversion attack would fail. Below are some thoughts that may help strengthen this aspect of the paper by putting more effort in this proof-testing TOFU:
- [Sanity check] Compute the analogous of Figure 3 (gradient inversion on the individual TOFU model updates) for the actual model updates (no TOFU involved). This important baseline would answer the question: is the used gradient inversion attack able to recover some original samples when we omit TOFU?
- [Push attack setup] Focus extreme setups vary the client batch size (smaller might be easier to attack) and number of iterations (smaller might be also easier to attack) and test extreme cases. For instance, in the single batch model update setting, does the gradient inversion attack still fail? Same question but in the (imaginary) case of single batch, single sample model update?
- [Redesign attack] Non-injectivity of the image $\mapsto$ gradient function. In other words, many (potentially non-image looking) samples might produce (approximately) the same gradient. This is why most of the attacks gradient reconstruction attacks guide the reconstruction with a prior on what the image is supposed to be like. Some other attacks might succeed here, such as [1] or [2]. Finally, one could imagine a novel attack in which the adversary knows this obfuscation method is being used (namely TOFU) to guide the reconstruction.

[1] "Gradient Inversion with Generative Image Prior", Jeon et al
[2] ""See through Gradients: Image Batch Recovery via GradInversion", Yin et al

---

> ### Author Response · Authors · 2023-02-10
> **Response to Reviewer v217, Part 1: Regarding Communication**
>
>
> We thank Reviewer v217 for an in-depth, valuable rebuttal. We sincerely believe that this rebuttal helps us improve our paper and are grateful to the reviewer. We address the concerns now and highlight the corresponding added portions in the revised version of the manuscript.
>
> &nbsp;
>
> ### COMMUNICATION
> 1. Compute cost for communication savings:
>
>     &nbsp;
>
>     1.1)  Cost of decoding: The computation for the server to decode the gradient just increases by one forward and backward pass per client, to decode the client’s gradient before aggregation. When all clients are synced (as in our setup), all the images from all the clients can be clubbed into a single batch to give the final average gradient in one forward pass.  The cost of decoding was in milliseconds and negligible to the overall cost.
>
>     &nbsp;
>
>     1.2) Cost of encoding: This is considerably more expensive, and we discuss it in detail. The two major parameters that control the cost of compute per communication rounds are number of iterations (Niter) and number of images in the synthetic dataset (Nimgs). We measure time taken to encode and decode the synthetic images in Section A.5, and tabulate it below. These experiments are performed on NVIDIA GeForce GTX 1060 GPU on VGG13 and the CIFAR-10 dataset for a Synfreq of 200 and a batchsize of 32. The training time for 200 minibatches is 5.8seconds.  The cost of encoding and decoding synthetic images:
>
> 	| Nimgs @Niter=1000 | Time (s) |
> 	|-------------------|----------|
> 	|         16        |   53.7   |
> 	|         32        |   84.1   |
> 	|         64        |   155.0  |
> 	|        128        |   293.2  |
>
> 	| Niter @Nimgs=32   | Time (s) |
> 	|-------------------|----------|
> 	|        250        |   22.5   |
> 	|        500        |   45.0   |
> 	|        1000       |   84.1   |
> 	|        2000       |   169.8  |
>
> 	The setting used will depend on the compute available and the accuracy drop acceptable in the application. While we report accuracy in the paper with Niter=1000, our experiments show good accuracies with Niter=500 as well (approximately 1% accuracy drop for CIFAR10, single device). At this setting, we add a compute cost of 9X the cost of creating only true gradients, in order to trade off 32x (from Table 5, for nimgs=32) lesser total parameters to communicate. The overall savings will depend on how much latency communication takes compared to compute.  We realize that this is a useful discussion that we only briefly touch upon it in the paper. We will be adding these details to the main paper, if accepted.
>
>  &nbsp;
>
>
> 2.  Full model updates: We only add a few rounds (3 for MNIST and 15 for CIFAR10) of full gradient exchange at the very end of learning. The single device accuracies shown in Table 1 do not use full gradient exchange, whereas Tables 7 and 8 show the results for both cases: with and without full gradient exchange. For Table 2 for example, the first set of columns correspond to the accuracy and communication efficiency achieved without any full gradient exchange rounds. The second column shows what happens if we exchange 3 (or 15) rounds of full gradient at the very end of learning (after synthetic gradients has achieved the accuracy shown in the previous set of columns) for MNIST (or CIFAR10). This shows that it is not just due to full gradient exchange that we are getting good accuracy, because full gradient exchange is only run for a fraction of the learning, and only after TOFU gets the accuracy to a good stable point. In all accuracy curves, the point where we start exchanging gradients instead of synthetic images is marked with a star indicated as switch2. Additionally, to show that loss curves follow accuracy curves, and TOFU does not cause a big increase in loss that full gradient exchange then recovers, we have added loss curves in Figure 9 in the appendix section A.3.4.

---

> ### Author Response · Authors · 2023-02-10
> **Response to Reviewer v217, Part 2: Regarding Privacy**
>
> ### PRIVACY
> &nbsp;
>
> 1.  Privacy clarifications: On a high level, our synthetic images produce an approximation of the gradients.  The approximation introduces noise that results in more communication rounds as shown in the results. However, as mentioned by the reviewer, we take advantage of the non-injectivity by weighing the gradients produced by synthetic images differently, soft-labels  and a severe reduction in the number of images to enforce noise, balanced by our optimization process to craft the noise to still be able to approximate the gradient as best as possible. We find the reviewers request of checks insightful, and we implemented these attacks and discuss our results here. We are unable to post images here unfortunately and have uploaded them to supplementary section A.4. We discuss the summary of those results here.
>
>     &nbsp;
>
> 	1.  Sanity check: In Figure 10, we show that the Inverting Gradients (IG) attack reveals image characteristics when applied to the true gradient but does not work on the synthetic gradient from TOFU for the same setup.
>
>     &nbsp;
>
> 	2.  Push attacks: We run TOFU for extreme versions and show results of gradient attacks. We varied the batchsize of the ground truth dataset in Figure 12, size of the synthetic dataset (Nimgs) in Figure 13 and frequency of synthesis (Synfreq) in Figure 14. The only case where TOFU fails is when both the batchsize and size of synthetic data are 1. As expected, IG is able to recover the true image from TOFU  since there is no role of spanning ratios or synthetic labels, and no condensation. But as we increase batchsize, Nimgs  or  Synfreq , we can see that the reconstruction fails. Our method relies on condensing larger amounts of information into smaller optimized image sets. and we recommend using our method with high synfreq to circumvent data leakage.
>
>     &nbsp;
>
> 	3.  Redesigning attacks and experimenting with more relevant attacks: IG is usually run with label knowledge, and not with the soft labels that our method uses. Hence, we adapt this attack to use the label as the maximum predicted probability of the synthetic images and then perform IG and show that this and the original method both fail to reconstruct data in Figure 11.  We also show the results of the 2 new attacks suggested by the reviewer, GIAS and GradInversion, alongside the results of DLG  and IG in Figure 15. We can see that the true gradients for this case leak data information when attacked, but TOFU’s reconstructed gradients do not. Additionally, we notice that all attacks (except DLG) converge to very similar looking noisy data. This is an additional sanity check to ensure that we are not just using bad hyperparameters for attacks.

---

### Review · Reviewer_TkVa · 2023-01-30

**Summary Of Contributions:**

The authors propose a way to improve communication efficiency in federated learning by learning synthetic data such that the synthetic data produce a gradient similar to that produced by true training data. Empirical evidence has shown that the method could compress the message being communicated while preserving the model utility.

**Audience:**

Yes

**Claims And Evidence:**

No

**Requested Changes:**

As stated in the weakness section, some of the changes I suggest includes:

- Adding citations and comparison to related works (e.g. [1,2,3]).
- Discuss the privacy aspect of this algorithm.
- Experiment in cross-device setting and with larger models. (A good example would be Reddit).
- Demonstrate the computation cost for the proposed method and compare that with prior works as well.

**Strengths And Weaknesses:**

Strengths:
- The empirical evidence seems solid.
- The idea of using synthetic data is interesting.

Weaknesses:

- This work seems to be very similar to this following work [1]. At first glance, the two works similarly use synthetic data for gradient compression. Comparison between the two works both intuitively and empirically is missing.
- Comparison with prior baselines on compression in federated learning is missing. To name a few baselines: Random Masking, Top-k Masking [3], Quantization [2], etc.
- If one of the motivation of this work is to protect BOTH communication efficiency and privacy, I found the privacy aspect of the proposed method largely unexplored in this work. For example, empirical evidence over how the method perform compared to FedAvg under common inversion attacks; theoretical evidence on how the proposed method could bound the Fisher Information, which could be viewed as an method to measure data leakage [4]. Discussion around these points would be helpful.
- Related to the privacy concern, it is unclear how does the differentially private version of the proposed method compared to the DP version of FedAvg, if privacy is one of the concern in this paper.
- How does the authors deal with the labels? Are the labels fixed or are the labels also trained?
- While the method significantly reduces the communication cost, it seems that the computation process is significantly more costly since it requires computing second order derivative (taking the gradient w.r.t the synthetic data, where the gradient it self is the cosine dissimilarity between two gradients). Could the authors discuss how what's the physical running time for this method.
- The scale of federated learning seems relatively small. Could the authors evaluate the method on cross-device settings, where the number of clients is large and one would need partial participation to train the model scalably.
- The method seems to work on larger models. I encourage the authors to evaluate the method on large NLP models, where the method could potentially gain more advantages.

[1] Hu, S., Goetz, J., Malik, K., Zhan, H., Liu, Z., & Liu, Y. (2022). Fedsynth: Gradient compression via synthetic data in federated learning. arXiv preprint arXiv:2204.01273.

[2] D. Alistarh, D. Grubic, J. Li, R. Tomioka, and M. Vojnovic. Qsgd: Communication-efficient sgd via gradient
quantization and encoding. Advances in Neural Information Processing Systems, 30, 2017.

[3]  A. Beznosikov, S. Horváth, P. Richtárik, and M. Safaryan. On biased compression for distributed learning. arXiv
preprint arXiv:2002.12410, 2020.

[4] Hannun, A., Guo, C., & van der Maaten, L. (2021, December). Measuring data leakage in machine-learning models with fisher information. In Uncertainty in Artificial Intelligence (pp. 760-770). PMLR.

---

> ### Author Response · Authors · 2023-02-10
> **Response to Reviewer TkVa: Part 1**
>
> We would like to thank the reviewer for the encouraging comments, and address some concerns below.
>
> &nbsp;
>
> 1.  We would like to thank the reviewer for directing us to a relevant paper. We will add FedSynth to our literature review and discuss major points for comparison here. There are two main conceptual differences between FedSynth and our method.
>
>     &nbsp;
>
>     1.1)  Our method encodes both upstream and downstream communication, essentially incurring 2x the error. We get comparable results on the MNIST dataset as reported. This is not a 1:1 comparison as the network and hyperparameters used are different but we only mention this to qualify our results as competitive to theirs.
>
>     &nbsp;
>
>
>     1.2)  The main difference however is that we optimize synthetic data to match the true gradient, whereas  FedSynth optimizes the data to reduce the loss. Fedsynth’s approach is very similar to DOSFL [1] instead of ours, and we discuss DOSFL in our literature review. Both these methods use a form of dataset condensation as their backbone, which differs from our concept of gradient matching. The main advantage we have over dataset condensation is that the latter is only shown to work for smaller datasets currently. The drop shown for CIFAR datasets is approximately 30\% accuracy, and hence we see both DOSFL and FedSynth show results only upto MNIST. However, for our case, from Table 2, it can be noted that we get an average of 3% accuracy drops for CIFAR10. However, it is still a very relevant  paper and we will updated this in the literature review along with the others mentioned, if accepted.
>
>
>     &nbsp;
>
> 2.  Regarding the weakness of the privacy claim. We have added many new experiments to our appendix to strengthen our claim in appendix section A.4. Here, we provide a summary of the new experiments to empirically reinforce the resulting data obfuscation from our method. We hope that these additional results count towards strengthening our claim on privacy.
>     2.1)  In Figure 10, we show that for the same settings, IG is able to recover user data when true gradients are shown, and fails to recover it when TOFU is used.
>     2.2)  In Figure 11, we show traditional IG attack and then modify it to work with soft labels. We show that both settings fail to reveal data.
>      2.3)  In Figure 12, we start with the very basic case of a single true image’s gradient encoded into a single fake image. In this case, IG is able to recover the true gradient as expected, since there is no condensation happening here, and no role for spanning ratios and soft labels. But as we increase batchsize in Figure 12, Nimgs in Figure 13, and Synfreq in Fig 14, TOFU’s privacy claim holds and IG fails to recover data.
>      2.4)  In Figure 15, we show 4 attacks in total: IG, DLG, GradInversion and GIAS. We show that these attacks recover user data on true gradients, and fail to recover data with synthetic gradients produced by TOFU. We also show that these attacks converge to the same noisy looking data, to show that it is indeed our method that induces data obfuscation rather than a failing of these attacks.
>
>     &nbsp;
>
> 3.  The reviewer has correctly noted that we trade off computation time for communication efficiency.   The two major parameters that control the cost of compute per communication rounds are number of iterations (Niter) and number of images in the synthetic dataset (Nimgs). We measure time taken to encode and decode the synthetic images in Section A.5, and tabulate it below. These experiments are performed on NVIDIA GeForce GTX 1060 GPU on VGG13 and the CIFAR-10 dataset for a Synfreq of 200 and a batchsize of 32. The training time for 200 minibatches is 5.8seconds.  The cost of encoding and decoding synthetic images:
>
> 	| Nimgs @Niter=1000 | Time (s) |
> 	|-------------------|----------|
> 	|         16        |   53.7   |
> 	|         32        |   84.1   |
> 	|         64        |   155.0  |
> 	|        128        |   293.2  |
>
> 	| Niter @Nimgs=32   | Time (s) |
> 	|-------------------|----------|
> 	|        250        |   22.5   |
> 	|        500        |   45.0   |
> 	|        1000       |   84.1   |
> 	|        2000       |   169.8  |
>
> 	The setting used will depend on the compute available and the accuracy drop acceptable in the application. While we report accuracy in the paper with Niter=1000, our experiments show good accuracies with Niter=500 as well (approximately 1% accuracy drop for CIFAR10, single device). At this setting, we add a compute cost of 9X the cost of creating only true gradients, in order to trade off 32x (from Table 5, for nimgs=32) lesser total parameters to communicate. The overall savings will depend on how much latency communication takes compared to compute.  We realize that this is useful discussion that we only briefly touch upon it in the paper. We will be adding these details to the main paper, if accepted.

---

> ### Author Response · Authors · 2023-02-10
> **Response to Reviewer TkVa: Part 2**
>
> 4.  How we deal with labels: We learn the labels as soft labels, along with the spanning ratios and images. We discuss it in more detail in section 3.2.
>
>     &nbsp;
>
> 5.  We thank the review’s advice of scaling to NLP datasets. NLP is definitely interesting and we are considering language models for future work. We hope that the good results we get on larger image datasets are of enough to qualify the merit of our paper for
>
> [1]  Zhou, Yanlin, et al. "Distilled one-shot federated learning." arXiv preprint arXiv:2009.07999 (2020).

---

### Decision · Action_Editors · 2023-03-06

**Recommendation:** Reject

**Comment:**

In agreement with the reviewers, I encourage the authors to rework the privacy statements in their paper, ideally by strengthening these statements in establishing rigorous resilience guarantees to substantial families of attacks.

**Audience:**

This paper's topic is a good match for the TMLR audience.

**Claims And Evidence:**

Some claims are supported by convincing and clear evidence, but as all reviewers noted, the claims of privacy protection are insufficiently supported, as no formal guarantees are established, only limited empirical evidence in the face of very specific attacks.